# Taking advantage of reference-guided assembly in a slowly-evolving lineage: Application to *Testudo graeca*

**Andrea Mira-Jover**[1], **Eva Graciá**[1]*, **Andrés Giménez**[1], **Uwe Fritz**[2], **Roberto Carlos Rodríguez-Caro**[3], **Yann Bourgeois**[4]*

**1** Ecology Area, University Institute for Agro-food and Agro-environmental Research and Innovation (CIAGRO), Miguel Hernández University, Elche, Carretera de Beniel, Orihuela (Alicante), Spain, **2** Museum of Zoology, Senckenberg Dresden, Dresden, Germany, **3** Departamento de Ecología, Universidad de Alicante, San Vicent del Raspeig, Spain, **4** DIADE, University of Montpellier, Montpellier, France

* egracia@umh.es (EG); yann.bourgeois@ird.fr (YB)

## Abstract

### Background

Obtaining *de novo* chromosome-level genome assemblies greatly enhances conservation and evolutionary biology studies. For many research teams, long-read sequencing technologies (that produce highly contiguous assemblies) remain unaffordable or unpractical. For the groups that display high synteny conservation, these limitations can be overcome by a reference-guided assembly using a close relative genome. Among chelonians, tortoises (Testudinidae) are considered one of the most endangered taxa, which calls for more genomic resources. Here we make the most of high synteny conservation in chelonians to produce the first chromosome-level genome assembly of the genus *Testudo* with one of the most iconic tortoise species in the Mediterranean basin: *Testudo graeca*.

### Results

We used high-quality, paired-end Illumina sequences to build a reference-guided assembly with the chromosome-level reference of *Gopherus evgoodei*. We reconstructed a 2.29 Gb haploid genome with a scaffold N50 of 107.598 Mb and 5.37% gaps. We sequenced 25,998 protein-coding genes, and identified 41.2% of the assembly as repeats. Demographic history reconstruction based on the genome revealed two events (population decline and recovery) that were consistent with previously suggested phylogeographic patterns for the species. This outlines the value of such reference-guided assemblies for phylogeographic studies.

### Conclusions

Our results highlight the value of using close relatives to produce *de novo* draft assemblies in species where such resources are unavailable. Our annotated genome of *T. graeca* paves the way to delve deeper into the species' evolutionary history and provides

**Data Availability Statement:** ***PA AT ACCEPT: Please check if data has been inputted in the NHI-NCB as links are currently empty on the database at RTC***"Yes - all data are fully available without

restriction" "All genomic and sequence files are available from the NHI-NCBI BioProject database (accession number PRJNA1086345). All scripts used for this study are freely available at https://github.com/YannBourgeois/Scripts_Genome_assembly_Tgraeca

**Funding:** This work was supported by Project PID2019-105682RA-I00 and TED2021-130381B-I00, funded by the Spanish Ministry of Innovation, Science and Universities (MICIU/AEI/10.13039/501100011033), the last also with the support of the European Union "NextGenerationEU"/PRTR". Roberto Carlos Rodríguez Caro was supported by European Union-Next Generation EU in the Maria Zambrano Programme (ZAMBRANO 21-26).

**Competing interests:** The authors have declared that no competing interests exist.

a valuable resource to enhance direct conservation efforts on their threatened populations.

## Introduction

Whole genome sequencing (WGS) has become a powerful tool in evolutionary and conservation biology due to the progressive reduction of practical and economical efforts to generate genomic libraries [1]. New high-throughput sequencing methods can be used to produce highly contiguous reference genomes for non-model or "obscure" organisms [1, 2]. Long-read DNA sequencing (e.g., Oxford Nanopore Technologies or PacBio) is a promising technique to generate high-quality reference genomes and is established as the future of *de novo* assemblies [3–6]. However, long read technologies remain expensive for species with large genomes, and require large amounts of high-molecular-weight DNA to be efficient [6]. The involved extraction protocols require fresh or flash-frozen tissues that cannot always be obtained for many laboratories and study systems [3, 5, 6]. Meanwhile short-read techniques remain cheaper and easier to use than long-read methods, and allow for using degraded samples [4–6]. Mapping-based and reference-guided assemblies (alignment of contigs/scaffolds to a close relative reference genome) provide a powerful tool to generate contiguous genomes using short reads [4]. The long scaffolds obtained from the close relative can be used to anchor the typically short contigs obtained with short reads [4, 7]. This method is particularly interesting for highly conserved syntenic genomes or even for extinct species, where mapping back the reference genome against extant relatives is the only option [4, 7–10]. Reference-guided assemblies provide an interesting feedback loop: the more high-quality reference genomes are obtained, the more likely it is that a close relative of the species of interest will be available [7, 9].

Chelonians, the vertebrate group that includes tortoises and turtles, is remarkable for its highly conserved synteny [11, 12]. The very wide diversity of ecological niches of chelonians is not reflected in their functional diversity or genome organization, which remains highly conserved across taxa while nucleotide divergence is particularly reduced and considered a slowly-evolving group (except for mitogenomes) [11–15]. High synteny is often observed in reptiles, including birds, while mammals tend to display more dynamic genomes (for a review see [16]). High synteny may be facilitated by long generation times typically found in chelonians, reducing the odds of rearrangements during meiosis. The lack of recently active transposable elements (TEs) may also contribute to prevent abnormal recombination between distant copies [17].

Genetic and genomic resources for chelonians have increased over the past few years. For example, a phylogeny obtained from 98 mitogenomes has contributed to address how ancestral extinctions, niche diversity and biogeography have impacted extant diversity [18]. However, microevolutionary processes like gene flow, genomic recombination, introgression, or hybridization, cannot be extensively addressed using only mitogenome trees [18]. Nuclear reference genomes are also becoming increasingly available. There are currently 38 reference genomes from 14 different families in the NCBI database (Table 1). These genomic resources have shed light on speciation events, ancient and recent demographic changes, and are also promising for addressing future studies, such as genomic determinants of aging, immunology, aridity tolerance, or gigantism in chelonians [12–14, 19–23]. However, the majority of these reference genomes represent aquatic species, particularly freshwater turtles (Table 1). Land tortoises or simply tortoises (Testudinidae) are the most threatened family of all chelonians [24], but only five annotated testudinid genomes have been published, with three from the

**Table 1. State of the art of reference genome availability for chelonians.** Species and subspecies are classified and divided into the main ecotypes (freshwater, marine, or terrestrial species). Estimated genome size is represented as Gb for all species. Assembly level represents the highest level for any object in the assembly (i.e., the sequence organization or connection among them). Sequencing technology is defined for every species jointly with N50, which indicates if the genome was sequenced using long-read (expressed on Mb) or short-read methods (expressed on kb).

| Type | Species/subspecies | Genome size (Gb) | Assembly level | Sequencing technology | N50 | NCBI Accession (GenBank) |
|------|--------------------|------------------|----------------|-----------------------|-----|--------------------------|
| Freshwater | *Actinemys marmorata* | 2.30 | Scaffold | PacBio Sequel II; Illumina NovaSeq; Dovetail OmniC | 75.1 Mb | GCA_022086475.1 |
| | *Actinemys pallida* | 2.33 | Scaffold | PacBio Sequel II and Sequel IIe; Dovetails OmniC; Illumina NovaSeq | 94 Mb | GCA_023634205.1 |
| | *Apalone spinifera* | 1.90 | Chromosome | Illumina HiSeq | 14.7 kb | GCA_030068395.1 |
| | *Carettochelys insculpta* | 2.18 | Chromosome | PacBio Sequel | 126.5 Mb | GCA_033958435.1 |
| | *Chelydra serpentina* | 2.26 | Scaffold | PacBio Revio | 47.4 Mb | GCA_018859375.1 |
| | *Chrysemys picta bellii* | 2.48 | Chromosome | 454 Life Sciences | 21.3 kb | GCA_000241765.5 |
| | *Cuora amboinensis* | 2.21 | Scaffold | Illumina | 47.2 kb | GCA_004028625.2 |
| | *Cuora mccordi* | 2.39 | Scaffold | 10X Genomics | 74.3 kb | GCA_003846335.1 |
| | *Dermatemys mawii* | 1.87 | Scaffold | 10X Genomics Chromium | 180 kb | GCA_007922305.1 |
| | *Emydoidea blandingii* | 2.3 | Scaffold | PacBio Sequel; Illumina NovaSeq | 43.1 Mb | GCA_036785055.1 |
| | *Emydura macquarii macquarii* | 1.92 | Contig | Oxford Nanopore PromethION; Illumina HiSeq | 17.1 Mb | GCA_026122565.1 |
| | *Emydura subglobosa* | 1.99 | Scaffold | 10X Genomics Chromium | 351 kb | GCA_007922225.1 |
| | *Emys orbicularis* | 2.31 | Chromosome | PacBio Sequel II HiFi; Bionano DLS; Arima Hi-C v2 | 91.3 Mb | GCA_028017835.1 |
| | *Glyptemys insculpta* | 2.32 | Scaffold | PacBio Sequel; Illumina NovaSeq | 95.7 Mb | GCA_032172135.1 |
| | *Graptemys geographica* | 2.3 | Contig | PacBio Revio | 107 Mb | GCA_037349215.1 |
| | *Macrochelys suwanniensis* | 2.13 | Chromosome | PacBio Sequel II HiFi; Arima Hi-C v2 | 43.9 Mb | GCA_033296515.1 |
| | *Malaclemys terrapin pileata* | 2.21 | Chromosome | PacBio Sequel II HiFi; Bionano Genomics DLS; Arima Hi-C v2 | 75.6 Mb | GCA_027887155.1 |
| | *Mauremys mutica* | 2.48 | Chromosome | PacBio | 15 Mb | GCA_020497125.1 |
| | *Mauremys reevesii* | 2.37 | Chromosome | Oxford Nanopore; Illumina | 33.4 Mb | GCA_016161935.1 |
| | *Mesoclemmys tuberculata* | 2.03 | Scaffold | 10X Genomics Chromium | 146.4 kb | GCA_007922155.1 |
| | *Pelochelys cantorii* | 2.16 | Chromosome | PacBio Sequel | 41.4 Mb | GCA_032595735.1 |
| | *Pelodiscus sinensis* | 2.20 | Scaffold | Illumina HiSeq 2000 | 22 kb | GCA_000230535.1 |
| | *Pelusios castaneus* | 2.04 | Scaffold | 10X Genomics Chromium | 74.9 kb | GCA_007922175.1 |
| | *Platysternon megacephalum* | 2.32 | Scaffold | Illumina | 213.6 kb | GCA_003942145.1 |
| | *Podocnemis expansa* | 2.45 | Scaffold | 10X Genomics Chromium | 134.8 kb | GCA_007922195.1 |
| | *Rafetus swinhoei* | 2.24 | Chromosome | Oxford Nanopore PromethION | 31 Mb | GCA_019425775.1 |
| | *Sternotherus odoratus* | 1.76 | Scaffold | PacBio Sequel; Illumina NovaSeq | 17 Mb | GCA_032164245.1 |
| | *Terrapene carolina triunguis* | 2.57 | Scaffold | 10X Genomics Chromium | 76.6 kb | GCA_002925995.2 |
| | *Trachemys scripta elegans* | 2.13 | Chromosome | Illumina NovaSeq; PacBio | 140 Mb | GCA_013100865.1 |

*(Continued)*

**Table 1.** (Continued)

| Type | Species/subspecies | Genome size (Gb) | Assembly level | Sequencing technology | N50 | NCBI Accession (GenBank) |
|---|---|---|---|---|---|---|
| Marine | *Caretta caretta* | 2.13 | Chromosome | Illumina NovaSeq; Oxford Nanopore PromethION | 18.2 Mb | GCA_023653815.1 |
| | *Chelonia mydas* | 2.13 | Chromosome | PacBio Sequel I CLR; Illumina NovaSeq; Arima Genomics Hi-C; Bionano Genomics DLS | 39.4 Mb | GCA_015237465.2 |
| | *Dermochelys coriacea* | 2.16 | Chromosome | PacBio Sequel I CLR; llumina NovaSeq; Arima Genomics Hi-C; Bionano Genomics DLS | 7 Mb | GCA_009764565.3 |
| | *Eretmochelys imbricata* | 2.30 | Chromosome | PacBio Sequel | 82 Mb | GCA_030012505.1 |
| Terrestrial | *Aldabrachelys gigantea* | 2.37 | Chromosome | PacBio Sequel | 58.7 Mb | GCA_026122505.1 |
| | *Chelonoidis niger abingdonii* | 2.30 | Scaffold | Illumina HiSeq; PacBio | 73.2 kb | GCA_003597395.1 |
| | *Gopherus agassizii* | 2.18 | Scaffold | Illumina HiSeq | 43.7 kb | GCA_002896415.1 |
| | *Gopherus evgoodei* | 2.30 | Chromosome | PacBio Sequel I; 10X Genomics linked reads; Arima Genomics Hi-C; Bionano Genomics DLS | 13 Mb | GCA_007399415.1 |
| | *Gopherus flavomarginatus* | 2.46 | Chromosome | PacBio Sequel I CLR; Bionano Genomics DLS; Arima Genomics Hi-C; 10X Genomics linked reads | 6.9 Mb | GCA_025201925.1 |

same genus (North American desert tortoises), *Gopherus flavomarginatus*, *G. evgoodei*, and *G. agassizii*. The other two represent giant tortoises from Galapagos (*Chelonoidis niger abingdonii*) and Aldabra (*Aldabrachelys gigantea*) [13, 14, 22] (Table 1). Of them, only three assemblies are annotated at the chromosome level: *G. flavomarginatus*, *G. evgoodei*, and *A. gigantea* (Table 1), all of them using long-read techniques.

In the Testudinidae family, the genus *Testudo* comprises five species of Mediterranean tortoises [24–27], and three of them are listed as threatened by the IUCN: *Testudo graeca* and *T. horsfieldii* are considered vulnerable [VU], and *T. kleinmanni* is listed as Critically Endangered [CE]) [24]. The spur-thighed tortoise (*T. graeca Linnaeus*, 1758) is the most widespread *Testudo* species in the Western Palearctic and shows an intricate phylogeographic history. Eleven mitochondrial lineages are described for *T. graeca*, and are divided into two different groups. The first, the eastern group, spans through the Near and Middle East and southeastern Europe, and consists of *T. g. ibera*, *T. g. terrestris*, *T. g. buxtoni*, *T. g. zarudnyi*, and *T. g. armeniaca* [27]. The second, the western group, primarily inhabits North Africa, but also includes some isolated populations in southwestern Europe. It is represented by *T. g. graeca*, *T. g. whitei*, *T. g. marokkensis*, *T. g. nabeulensis*, *T. g. cyrenaica*, and an additional lineage awaiting its formal description [27]. Fossil-calibrated molecular clock analyses based on mitochondrial data suggest that the western group diverged from its sister taxon, *T. g. armeniaca*, during the Pliocene (7.95–3.48 Mya). Two independent diversification bursts took place during the Mio-Pliocene (8–2 Mya) for the eastern lineages, and during the Pleistocene (1–0.1 Mya) for the subspecies distributed in North Africa [27]. Southwestern European populations have their origin in North Africa [26] being historically introduced on Mallorca, Sardinia, and the Doñana National Park [27, 28]. An exception is a *T. g. whitei* population in southeastern Spain, with molecular markers indicating a range expansion from North Africa during the Late Pleistocene (20 kya) and subsequent natural expansion in southeastern Spain [26].

In the face of the conservation status of the *Testudo* species (and all other turtles and tortoises, with more than 50% of its species considered as Threatened [24]) and the singularity of the phylogeographic history of *T. graeca* throughout the Mediterranean, a reference genome for *Testudo* will greatly contribute to all future studies aimed at the conservation and better understanding of tortoises.

To address the lack of genomic resources for this genus, we present the first high-quality genome for *T. graeca*. We sequenced it using short-read technology on an Illumina platform to generate a draft assembly, and used an available reference genome of a close relative (*G. evgoodei*, diverged approximately 50 Mya) [18, 29] to scaffold and annotate the genome. Our work demonstrates the efficiency of the reference-guided assembly to create accurate *de novo* reference genomes that can serve for future studies.

## Material and methods

### Sample collection and sequencing

To sequence the whole genome of *T. graeca*, we sampled a fresh road-killed male tortoise from Murcia (southeastern Spain) (S1 Map of sample location). Both field sampling, and the collection and treatment of biological samples, were supported by the government of Murcia Region (AUF20140057) and Project Evaluation Agency of the Research Vice-Rectorate of Miguel Hernandez University (Elx, Spain) (UMH.DBA.EGM.03.19). The sample was stored at -18˚C. Tissues were extracted under sterile conditions and kept frozen until processing. DNA was extracted from muscle using the E.Z.N.A Tissue DNA kit (Omega Biotek) and eluting it in 100 μL. DNA quantification was performed by a Qubit High Sensitivity dsDNA Assay (Thermo Fisher Scientific) at a final concentration of 37.6 ng/μL. Genomic DNA libraries were constructed using the TruSeq Nano DNA kit and quality-checked in the TapeStation D1000 ScreenTape System (Agilent Technologies). Genomic libraries were sequenced on an Illumina NovaSeq with PE150 (paired-end) to obtain a total output of 220 Gb (*c*. 100X depth of coverage). Raw FASTQ files were quality-checked using FastQC v0.11.5 [30] (S1 Appendix). All the procedures were carried out by AllGenetics & Biology S.L. following its company protocols.

### Genome assembly

Before starting the assembly, we adapter-trimmed all sequences and quality-filtered them using Trimmomatic 0.39 [31]. We discarded reads with a Phred quality score lower than 28 and trimmed reads when quality dropped below 5. We removed the Illumina adapters (TruSeq3-PE) and discarded reads shorter than 40 bp. Overlapping reads were merged employing Pear v0.9.11 (default overlap of 10 bp) [32]. Sequencing errors were corrected using SOAPec v. 2.0.3 by specifying k-mer size as 27, and the cut-off size as 3, for removing low-frequency k-mers. Assembly was performed using SOAPdenovo2 (version 2.04 release 242) [33] with a range of increasing k-mer values (27, 37, 47, 57, 67, 77, 87, 97, 107). We also tested using k-mer sizes (121 and 127 bp), predicted as being optimal by KmerGenie 1.7051 [34]. KmerGenie was also deployed to predict genome size.

The assemblies that employ short reads are generally fragmented and consist in thousands of short contigs. To improve our draft assemblies, we used ntJoin [4] to scaffold our draft assemblies with *G. evgoodei* as a reference given its high-quality chromosome-level assembly with a few unplaced scaffolds. We ran ntJoin with a range of word sizes (100 bp, 250 bp, 500 bp and 1000 bp) and a set of k-mer values (16, 24, 32, 40, 48, 56, 64). Any gaps between contigs were then closed using GapCloser v1.12 [33]. To confirm the efficiency of this approach, we examined the completeness of our assembly with BUSCO (BUSCO score v 5.3.0) [35]. We tested the continuity and the presence of 5310 shared genes of tetrapods (tetrapoda_0db10, from OrthoDB database) before and after applying ntJoin, and always after gap removal with GapCloser. We also compared the quality of the different assemblies by examining N50, L50 and other statistics using the stats.sh script from the bbmap suite (BBTOOLS 38.18). Quality criteria were assessed according to the percentage of gaps, the number of the longest scaffolds

covering half the assembly (L50), and the shortest length of those scaffolds (N50). Therefore, we retained the assembly with the lowest gap percentage, the highest N50 and the lowest L50.

## Repeat analysis

For identifying repeated elements, we used RepeatModeler v2.0.2 [36] to create *de novo* predictions of repetitive sequences and to construct a library of repetitive elements for *T. graeca*. To mask the genome, we combined this *de novo* annotation with an existing consensus of repetitive sequences for tetrapods using the freely available resources (Dfam) provided with RepeatMasker v4.1.2 [36]. We ran the latter program with RMBLAST v2.11.0 to classify and annotate all the repeat families. Then we built a Repeat Landscape to compare *T. graeca* repeat content to other species. We explored the age distribution of TEs by examining the divergence among the different TE families with the calcDivergenceFromAlign.pl script from the RepeatMasker package.

## Gene annotation

Gene finding was performed using BRAKER2 v2.1.6 [37], which incorporates a combination of tools to predict gene coordinates and generates gene structure annotations [37–39]. As we do not have access to the RNAseq data for our species, we applied the BRAKER pipeline using the "C" option to incorporate "proteins of any evolutionary distance" into our target species. Because these methods work better with proteins from related species, we combined the protein annotations available for *G. evgoodei*, *A. gigantea*, and *Gallus gallus* with a set of vertebrate protein data obtained from OrthoDB (tetrapoda_odb10) using DIAMOND [40] to remove any redundant genes between both sources. We ran gene predictions on our masked genome to avoid wrongly annotating TEs as genes. Briefly, the pipeline involves running ProtHint [41] to generate hints of protein prediction by identifying alignments with sequences from close or distant relatives for *T. graeca* in the provided protein database. Annotation is further improved by training AUGUSTUS [38, 40, 41] on the set of hints to obtain the coordinates and predictions of introns, exons, and start/stop codons. We obtained Gene Ontology (GO) terms and gene names for the predicted genes with EggNOG-mapper v2 [42], and by a high-precision search among orthologous groups.

We also transferred the *G. evgoodei* annotation to the *T. graeca* draft genome using Liftoff with default parameters [43].

## Mitogenome reconstruction

We used MitoZ [44] with default parameters on a subset of 10 million pairs of reads to reconstruct the mitogenome of *T. graeca*. Several k-mer values were tested (59, 79, 99, 119, 141). The final assembly was obtained with a k-mer value 141. We aligned our mitogenome reference to other *Testudo* mitogenomes using MAFFT online with default parameters (https://mafft.cbrc.jp/). To further confirm the quality of our sequence, we checked its position in the phylogeny of complete *T. graeca* mitogenomes with a mitogenome from *Testudo marginata* as an outgroup (NCBI accession DQ080047.1). We also employed MAFFT online to run a Neighbor-Joining phylogenetic analysis on the alignment using 100 bootstrap replicates to calculate node support.

## Demographic history inference

Historical changes in the effective population size were inferred with the MSMC2 v2.1.4 software [45]. MSMC2 uses a Hidden Markov Model to estimate the most recent time since coalescence among the haplotypes under recombination. The method can be applied to a single

diploid genome, but requires heterozygous sites to be identified to obtain coalescence times between the two haplotypes. To do so, we realigned the reads on the reference genome using BWA-MEM-2 [46]. We ran freebayes v1.3.2 [47] to call variants from the generated alignment file (in BAM format). We restricted the analysis to the nine longest scaffolds. To identify poorly mappable regions, we employed GenMap v1.3.0 [48] on the genome assembly. We used BEDTOOLS v2.29.2 [49] to obtain the depth of coverage along the genome (average of 100x) from the BAM file. We masked the regions with a mappability lower than 1 and a depth of coverage below 10X. With VCFtools v0.1.16 [50], we filtered the SNP variants from each chromosome and excluded the sites with a genotype quality lower than 30 and depth less than 10X, or more than 200X. Using the *generate_multihetsep.py* script (provided by msmc-tools, a repository containing utilities for MSMC2, https://github.com/stschiff/msmc-tools), we merged VCF outputs and mask files together to generate the input files for MSMC2. The software was run with default parameters by defining time segmentation as '*-p 1*2+25*1+1*2 +1*3*' and grouping the first and last two-time intervals to force the coalescent rate to remain constant.

The coalescence rates estimated by MSMC2 were converted into generations at a mutation rate of 6 x $10^{-10}$ bp/year based on the *c.* 6% divergence between the *G. evgoodei* and *T. graeca* genomes, which diverged *c.* 50 Mya (substitution rate of 3% per lineage over 50 My, or 6 x $10^{-10}$ substitutions per year) [29]. Generation time was estimated at 17.72 years, as in Graciá et al. [26].

## Results

### Genome sequencing and assembly

For whole genome sequencing, we generated a total of 2 x 913,404,107 high-quality paired-end short Illumina reads with an average sequence length of 151 bp and a GC content of 45%. After adapter and low-quality trimming, we conserved 872,311,739 sequenced reads. *Kmer-Genie* predicted an optimal k-mer value for the genome *de novo* assembly of 125 bp for an estimated genome size of 2,172,882,866 bp. This estimate is consistent with the sizes obtained for other chelonian genomes assembled at the chromosome level (ranging from 2.13 Gb for *Chelonia mydas* and 2.48 Gb for *Chrysemys picta bellii*). Assembling with SOAPdenovo and a k-mer size of 87, produced the assembly with the highest scaffold and contig L50 (5.67 kb and 4.01 kb, respectively; see also S1 Fig). This assembly was used for further scaffolding employing ntJoin. A word size of 100 and a k-mer size of 24 resulted in the reference-guided assembly with the highest contig N50 value (3.6 kb) and the smallest gap proportion (13.24%). The proportion of gaps dropped to 5.37% after running GapCloser (Table 2), but the contig N50 rose to 132,837 bp. This scenario suggests that a large proportion of contigs and scaffolds obtained by SOAPdenovo were correctly positioned in relation to one another to ensure efficient gaps filling in the reference-guided assembly (Table 2). The BUSCO complete score rose from 30.3% for the SOAPdenovo assembly to 96.7% after scaffolding with ntJoin and gaps filling, while the proportion of the fragmented and missing genes dropped from 28.8% to 1.1% and from 40.9% to 2.2%, respectively (Table 2 and Fig 1A).

### Repeat content

The fraction of repetitive elements identified by RepeatModeler/RepeatMasker was 41.02% for a total of 956,680,633 bp. This falls in line with other related chelonian genomes (Table 3). Long interspersed nuclear elements (LINEs) were the most abundant class of repetitive elements (11.11%), followed by DNA transposons (7.19%), short interspersed nuclear elements (SINEs) (2.06%) and long terminal repeats retrotransposons (LTR-RTs) (3.11%). Unclassified elements accounted for 17.04% of the draft genome. Divergence of repeats from their

**Table 2. Comparison of the assembly statics and BUSCO analysis before and after correcting the draft assembly by the ntJoin method.**

| | SOAPdenovo2 assembly (k = 87) | Reference-guided assembly after running GapCloser |
|---|---|---|
| **Assembly statics** | | |
| *Main genome scaffold (L/N)* | 3,895,059/2,588.160 MB | 84,170/2,332.51 MB |
| *Main genome contig (L/N)* | 4,236,106/2,563.699 MB | 112,106/2,207.2 MB |
| *GAPs* | 0.945% | 5.37% |
| *Main genome scaffold L/N50* | 118,065/5,676 bp | 6/130.16 MB |
| *Main genome contig L/N50* | 157,154/4,014 bp | 4,605/132.84 KB |
| *Main genome scaffold L/N90* | 817,946/240 bp | 31/3.92 MB |
| *Main genome contig L/N90* | 1,228,070/203 bp | 21,136/13.46 KB |
| *Max scaffold length* | 220,437 bp | 348.492 MB |
| *Max contig length* | 220,066 bp | 1.21 MB |
| *Scaffolds > 50 kb* | 157 | 144 |
| *% Main genome in scaffolds >50 kb* | 0.39% | 92.15% |
| **BUSCO v5.3.0 search** | | |
| *Complete and single-copy* | 1561/29.4% | 5082/95.7% |
| *Complete and duplicated* | 49/0.9% | 55/1% |
| *Fragmented* | 1530/28.8% | 56/1.1% |
| *Missing* | 2170/40.9% | 117/2.2% |
| *Total groups searched* | 5310 | 5310 |

consensus sequences showed a mode at 7% (S2 Fig), which suggests limited activity of transposable elements (TEs) over recent (< 10 Mya) evolutionary times.

## Gene annotation

The *de novo* annotation of the genome obtained with BRAKER2 using a bank of vertebrate protein sequences recovered 24,397 genes with an average length of 6808 bp (Fig 1B). Although the total number of genes recovered is in line with the number estimated for other Testudinidae, their average length is one order of magnitude shorter than those of *G. evgoodei* or *A. gigantea* [14]. As the *G. evgoodei* annotation benefited from transcriptomic data, we transferred it to our own reference. Of the 19,808 coding genes, 462 could not be transferred to the *T. graeca* genome (Table 4). Most (70%) of the genes that were reconstructed *de novo* overlapped with a gene from the *G. evgoodei* annotation, which confirmed that our *de novo* annotation recovered the majority of coding genes, but likely not their full-length sequence.

## Mitogenome reconstruction and phylogenetic placement

We reconstructed the mitogenome using MitoZ, and obtained a 16,928-bp-long circularized sequence with 37 annotated mitochondrial genes (Fig 1C). Depth of coverage was homogeneous along the sequence (average depth +/- s.d.: 172X +/- 19 when excluding the outlier control region), except for the control region where it peaked at more than 2000X. The comparison to the other two complete *T. graeca* mitogenomes (NCBI accession numbers: DQ080049.1, DQ080050.1) confirmed the completeness of our assembly outside the control region, and the whole sequence was fully aligned to the other references with 96.2% and 98.7% identity. Our mitogenome was shorter than the other references (lengths: 17,674 and 19,278 bp) due to a shorter control region. This region could not be fully assembled due to its

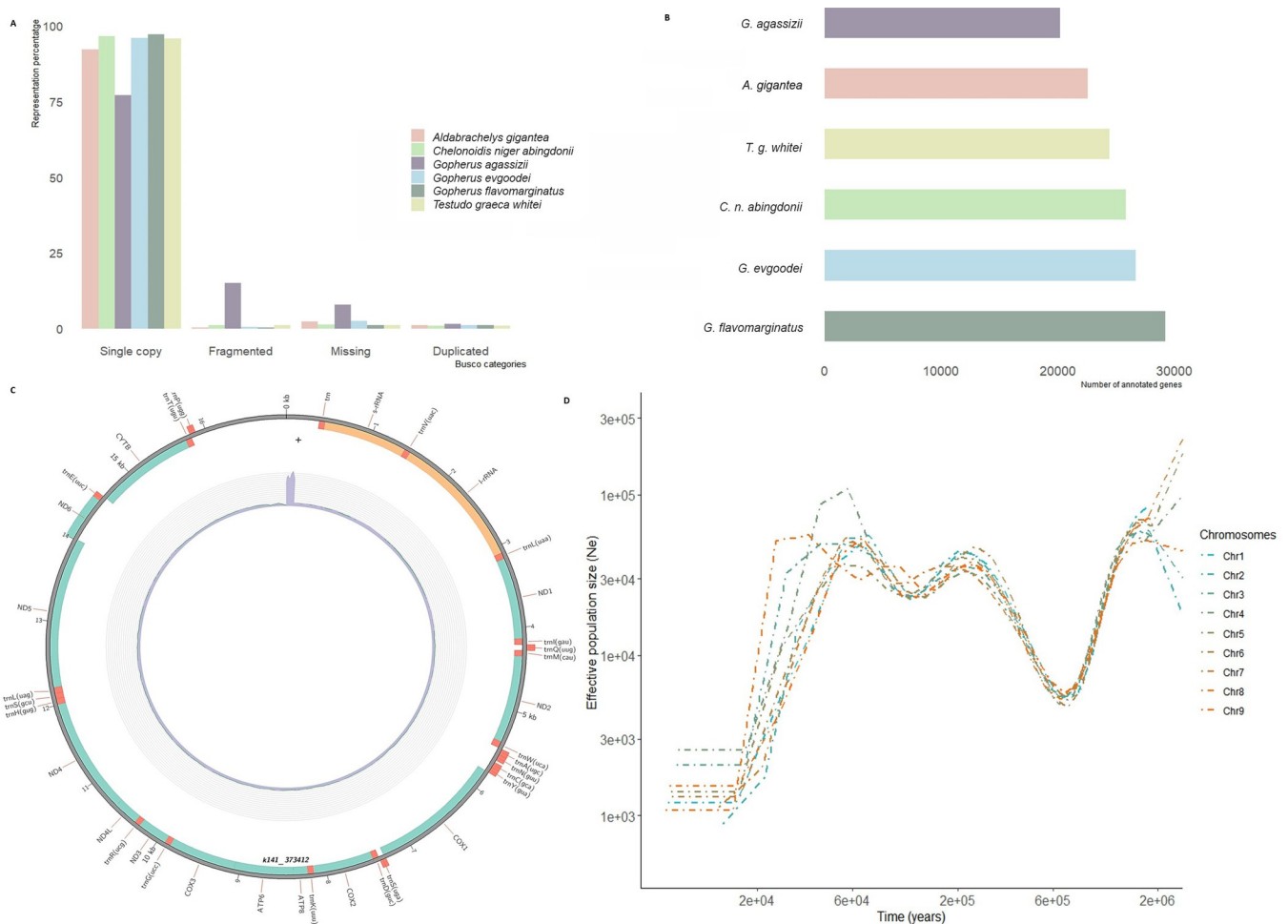

**Fig 1. Representation of the main genome assembly results.** (A) BUSCO completeness comparison with the five assembled Testudinidae genomes. (B) Number of genes annotated for the five Testudinidae genomes and for *T. graeca*. (C) Circos plot of the mitogenome, with the position of the annotated genes and depth of coverage (inner circle). (D) Demographic reconstruction of the large annotated chromosomes for *T. graeca*.

repetitiveness and the relatively short length of our reads and inserts. A Neighbor-Joining phylogenetic analysis places our reference close to the Tunisian sample with high support, but more distant from the Turkish sample. All this is consistent with expectations given the species' biogeography (S3 Fig).

## Demographic reconstruction

To evaluate the applicability of *T. graeca*'s reference genome for demographic analyses, we reconstructed its past demographic history using MSMC2 (Fig 1D). We estimated an effective population size ($N_e$) to have revolved around 30,000 individuals over the last 3 My, with two population decline events: the first one around 1 Mya, when $N_e$ declined to *c*. 6000 individuals; the second decline more recently occurred at 40–20 kya, with $N_e$ declining to *c*. 1000 individuals. Of these declines, we observed a significant recovery in $N_e$, approximately by one order of magnitude, during the period between 600–200 kya.

**Table 3. Representation of the different families of repetitive elements found in the *Testudo graeca* assembly and other closely related available chelonians (*Gopherus agassizii* and *Aldabrachelys gigantea*).** *Testudo graeca* and *A. gigantea* show similar rates of retroelements as DNA transposons over the genome, while the *G. agassizii* masked region results in a greater presence of retroelements, but fewer DNA transposons. We note that the *G. agassizii* study only ran RepeatModeler on unmasked regions, while the *T. graeca* and *A. gigantea* studies cover the whole genome. Unreported statistics are indicated as 'NA'.

| | Number of elements/Length occupied (bp) | | |
|---|---|---|---|
| | *T. graeca* | *G. agassizii* | *A. gigantea* |
| **Interspersed repeats** | 944,683,527 | NA | 1,087,548,019 |
| **Retroelements** | 1,296,773 / 379,529,682 | NA / 505,075,885 | 1,177,209 / 482,092,777 |
| *SINEs* | 331,628 / 47,948,990 | NA / 44,092,705 | 51,461,867 / 326,746 |
| *Penelope* | 139,606 / 30,016,241 | NA | NA |
| *LINEs* | 838,188 / 259,068,640 | NA / 276,159,275 | 293,395,900 / 695,701 |
| *LTR* | 126,957 / 72,512,052 | NA / 184,823,905 | 137,235,010 / 154,762 |
| **DNA transposons** | 661,863 / 167,700,840 | NA / 297,537,719 | 198,183,931 / 642,321 |
| **Unclassified** | 2,294,377 / 397,453,005 | NA / 203,226,010 | 407,271,311 / 1,902,917 |
| **Others** | | | |
| **Small RNA** | 48,814 / 9,288,904 | NA / 9,451,148 | NA |
| **Satellites** | 1,775 / 691,439 | NA / 1,402,596 | NA |
| **Simple repeats** | 216,453 / 8,510,043 | NA | NA |
| **Low complexity** | 33,453 | NA | NA |

## Discussion

Using Illumina NovaSeq PE150 sequencing, we generated the first high-quality draft genome assembly for *T. graeca*, including its mitogenome. The reference-guided assembly notably increased sequence contiguity and facilitated annotation. This illustrates the efficiency of the reference-guide assembly for chromosome-level scaffolding and gene annotation by providing a resource for comparing genome organization and diversity within and across clades [4, 51]. However, there are always potential inherent biases towards the reference and *de novo* assembled genome (mainly due to divergent regions between the chosen and target assemblies or errors in reference sequence annotation) [7]. In our case, taking *G. evgoodei* as a reference drastically reduced the gap contents and the presence of any "fragmented" and "missing genes". BUSCO completeness scored favorably with other chelonians (Fig 1A), but it should be noted that Çilingir et al. [14] conducted a BUSCO analysis using OrthoDB v10 datasets from phylum (vertebrata_odb10) and class (sauropsida_odb10) instead of all the tetrapods.

BRAKER2 gene prediction estimated a similar number of genes to other Testudinidae (Fig 1A and 1B). However, lack of RNA-seq data prevented us from obtaining full-length transcripts and genes. By making the most of the high contiguity of our reference, and combined with conserved synteny and high identity with *G. evgoodei*, we were able to transfer the annotation of the latter to the *T. graeca*'s genome (Table 4).

As *T. graeca* shows accurate differences between population and lineages, the genome herein produced is valuable for further population genomics studies. Using reference genomes

**Table 4. Comparison of the *de novo* annotation with BRAKER and Liftoff of the already existing annotation from the related *Gopherus evgoodei*.**

| Annotation | Number of exons | Number of coding exons | Number of genes | Number of private genes | Average gene length (+/- s.d.) | Average coding exon length (+/- s.d.) |
|---|---|---|---|---|---|---|
| **BRAKER** | 12,9168 | 12,9168 | 24,397 | 7293 | 6808 +/- 10,306 | 225 +/- 352 |
| **Liftoff from G. evgoodei** | 32,4881 | 31,8841 | 19,346 | 2242 | 38,670 +/- 67,033 | 195 +/- 307 |

from distantly related species can negatively impact SNP calling by underestimating the number of variants or biasing heterozygote calling [52]. This effect is significant in turtles and tortoises [12], and obtaining a reference from the same species ensures accurate future genotyping, thereby avoiding bias in the analyses. Highly contiguous genomes are also essential for proper gene annotation. This is clearly reflected by the drastic improvement in our BUSCO scores, with the proportion of complete single copies recovered rising from 29.4% to 95.7%. The average gene length of *c.* 38,000 bp in *Gopherus* is nearly one order of magnitude higher than the scaffold N50 of 5676 bp before reference-guided assembly. At last, contiguous reference genomes are critical for accurate population genetic inference. In humans, approaches such as MSMC2 and related methods lose in accuracy for scaffold lengths under 100 kb– 1 Mb [53]. Before reference-guided scaffolding, the longest scaffold of our assembly was 220,437 kb long, while only four scaffolds were longer than 100 kb. Even assuming human-like mutation and recombination rates, which are likely higher than in tortoises, the *T. graeca* reference would therefore have lacked of long enough scaffolds for MSMC-like approaches, impairing demographic reconstructions. Long scaffolds are also important for genome scans of selection, which rely on the lengths of haplotypes to detect possible selective sweeps [54].

Demographic history reconstructions can address biological questions and retrace the evolutionary dynamics underlying the current distribution and the population genetic status of species [45, 55]. The population size dynamics herein inferred aligns well with the past population changes proposed in previous studies that used mtDNA or microsatellites [26, 27]. The older population decline is compatible with the rapid radiation suggested for the North African lineages during the Pleistocene [27]. Today these subspecies show a clear niche differentiation in North Africa, particularly in relation to climate variables like rainfall [56]. The subsequent population recovery aligns with the diversification of *T. g. whitei*, and has been estimated to have occurred between 850 and 170 kya [27]. For this lineage, it has been suggested that it was confined to several refuge areas during the Last Glacial Maximum, from which it subsequently expanded [56], and a similar pattern can be anticipated during other glacial maxima. Repeated contractions and expansions may have greatly contributed to lineage diversification during the Pleistocene. Finally, the more recent decline is consistent with the bottleneck linked with the species' arrival in southeastern Spain, estimated to have occurred some 20 kya [26, 27].

As inferred for *T. g. whitei* in our demographic reconstruction, the currently available reference genome will increase our knowledge of the past population dynamics and other lineages' demographic history.

Our repetitive element analysis showed that *c.* 40% of the genome is made of TEs, which is a similar proportion to other chelonians, such as *Chelonia mydas*, *Chrysemys picta bellii*, or *Gopherus* spp., but lower than the estimates for *A. gigantea* (46.7%) or *Trachemys scripta elegans* (45%) (Table 2). The Interspersed Repeat Landscape suggests very low recent transposition given the observed age distribution of TEs. Recent TEs activity appears unlikely, and is possibly biased due to the difficulty of assembling highly repeated regions, and using reference-guided ones. However, this is unlikely given the Kmergenie estimates, which are consistent with the reconstructed genome length and consistent estimates with a c-value.

## Conclusion

In this study, we report the first reference genome for the genus *Testudo* and add *T. graeca*'s genome to the "toolkit" of genomic resources for tortoises. Given the shared synteny of Testudinidae, we made the best of the high-quality assemblies for another tortoise species, *G.*

*evgoodei*, to scaffold and annotate a chromosome-level assembly genome from short-read sequences. Thanks to this approach, we avoided the higher costs and sample quality challenges of long-read techniques, and make the most of the low error rate and the cost effectiveness of short read sequencing.

This newly generated reference genome will be useful for answering questions about the evolutionary history and conservation of the *T. graeca* complex, and possibly of other *Testudo* species.

## Supporting information

**S1 Map. Sample location coordinates.**
(PDF)

**S1 Appendix. FastQC report.**
(PDF)

**S1 Fig. Summary Statics of ntJoin scaffolding.** Upper row: summary statistics (scaffold, right, and contig L50, left) for SOAPdenovo2 assemblies using a range of k-mer size before scaffolding with ntJoin. The red dot indicates the assembly retained for the next step (scaffolding with ntJoin). Lower row: Percentage of gaps remaining after scaffolding the k = 87 SOAPdenovo2 assembly with ntJoin for k-mer size (right) and word size (left). The red dot indicates the assembly retained for the next step (GapCloser).
(PDF)

**S2 Fig. Repeated Landscape of *Testudo graeca*.** Kimura divergence of each repetitive element copy from its consensus is displayed as a barplot.
(PDF)

**S3 Fig. Mitochondrial phylogeny.** Phylogeny of full-length *Testudo graeca* mitochondrial sequences using *T. marginata* as an outgroup. Bootstrap support is indicated at nodes.
(PDF)

## Acknowledgments

We thank AllGenetics SL sequencing services, Portsmouth University for computational resources and to all the Ecology Area of Miguel Hernández University (specially to Paco Botella), Serbal, Ecologistas en Acción de Murcia, and Andalucía and Murcia Region governments for their field support. Finally, we acknowledge all projects who provided new and high-quality reference genomes of chelonians. The genome assembly was done on the Sciama High Performance Compute (HPC) cluster which is supported by the ICG, SEPNet and the University of Portsmouth.

## Author Contributions

**Conceptualization:** Eva Graciá, Yann Bourgeois.

**Formal analysis:** Andrea Mira-Jover, Yann Bourgeois.

**Funding acquisition:** Eva Graciá, Andrés Giménez.

**Resources:** Eva Graciá, Andrés Giménez, Roberto Carlos Rodríguez-Caro, Yann Bourgeois.

**Supervision:** Eva Graciá, Andrés Giménez, Yann Bourgeois.

**Validation:** Uwe Fritz, Roberto Carlos Rodríguez-Caro.

**Visualization:** Andrea Mira-Jover, Eva Graciá, Andrés Giménez, Uwe Fritz, Roberto Carlos Rodríguez-Caro, Yann Bourgeois.

**Writing – original draft:** Andrea Mira-Jover, Yann Bourgeois.

**Writing – review & editing:** Andrea Mira-Jover, Eva Graciá, Andrés Giménez, Uwe Fritz, Roberto Carlos Rodríguez-Caro, Yann Bourgeois.

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
