## [Decision Letter · Decision Letter 0]

13 May 2024

PONE-D-24-16548Taking advantage of reference-guided assembly in a slowly-evolving lineage: application to Testudo graeca.PLOS ONE

Dear Dr. Bourgeois,

Thank you for submitting your manuscript to PLOS ONE. After careful consideration, we feel that it has merit but does not fully meet PLOS ONE’s publication criteria as it currently stands. Therefore, we invite you to submit a revised version of the manuscript that addresses the points raised during the review process.

Although the manuscript is written well and the manuscript's goal is to demonstrate the benefits of using a reference genome, thus I believe the results should illustrate how much improvement there is when comparing reference-guided vs. de novo assemblies. I encourage authors to take note of the feedback provided by reviewers and to correct or revise their manuscript where required.

We look forward to receiving your revised manuscript.

Kind regards,

Murtada D. Naser

Academic Editor

PLOS ONE

Journal Requirements:

"This work was supported by Project PID2019-105682RA-I00 and TED2021-130381B-I00, funded by the Spanish Ministry of Science and Innovation (State Research Agency) (MICIU/AEI/10.13039/501100011033), the last also with the support of the European Union “NextGenerationEU”/PRTR”. "

"This work was supported by Project PID2019-105682RA-I00 and TED2021-130381B-I00, funded by the Spanish Ministry of Science and Innovation (State Research Agency) (MICIU/AEI/10.13039/501100011033), the last also with the support of the European Union “NextGenerationEU”/PRTR”."

"This work was supported by Project PID2019-105682RA-I00 and TED2021-130381B-I00, funded by the Spanish Ministry of Science and Innovation (State Research Agency) (MICIU/AEI/10.13039/501100011033), the last also with the support of the European Union “NextGenerationEU”/PRTR”."    

Reviewers' comments:

Reviewer's Responses to Questions

**Comments to the Author**

1. Is the manuscript technically sound, and do the data support the conclusions?

Reviewer #1: Partly

Reviewer #2: Partly

Reviewer #3: Yes

2. Has the statistical analysis been performed appropriately and rigorously? 

Reviewer #1: Yes

Reviewer #2: No

Reviewer #3: Yes

3. Have the authors made all data underlying the findings in their manuscript fully available?

Reviewer #1: Yes

Reviewer #2: No

Reviewer #3: No

4. Is the manuscript presented in an intelligible fashion and written in standard English?

Reviewer #1: Yes

Reviewer #2: Yes

Reviewer #3: Yes

5. Review Comments to the Author

Reviewer #1: In this manuscript, the researchers assembled a chromosomal-level genome for an iconic land tortoise species. With only cheap short-read sequencing data, the study took advantage of the highly conserved synteny in the Chelonian group to build a reference-guided assembly.

I have one general question. The paper intends to show the advantage of using a reference genome, so I think the result should show how much improvement there is when comparing reference-guided vs. de novo assemblies. It could be the BUSCO completeness, the number of annotated genes, or the demographic reconstruction difference. Other than this, I found most of the manuscript is written clearly except in a few places that need minor editing.

Page 2 Line 51: “valuable resource to increase making direct conservation efforts” should be rephrased, maybe “valuable resource for conservation efforts.”

Page 3 Line 56: to generate and sequence genomic libraries

Page 4 Line 80: here, readers would need more background on the highly conserved synteny, like comparing Chelonians to other groups of similar age.

Page 4 Line 85: “using only mitogenomes” or “using only the mitogenomes’ tree”

Page 4 Line 89: “addressing novel studies”? means “studying the origin of novel/adaptive traits” ?

Page 6 Line 119-120: I don’t understand the sentence starting with “In general.”

Page 6 line 124: I think there should be a sentence or two about the importance of high-quality genome for this group.

Page 7 Line 127: need some information on the “close relative”, like the divergence time between the two species

Page 7 Line 136: minus four degrees? Just need to confirm because usually the fridge is 4 degrees, and the freezer is -20

Page 8 Line 164: reference for the tetrapoda-0db10. Is it from the OrthoDB database?

Page 9 Line 177: TEs

Page 13 Line 265: “Over recent (>10Mya) evolutionary times” should be “(<10Mya)”?

Page 13 Line 265: I don’t understand why the same data needs a table and a figure. Also, why does A. gigantea have nothing in the “Others”?

Figure 1: missing ABCD, and the legend b) should be “the number of genes annotated.”

Reviewer #2: Comments to authors:

This manuscript adds considerable information to understanding full genomes of Testudines. The only major weakness is an incomplete description of how a demographic analysis can be undertaken from a single genome. The paragraph in the Results that is related to this analysis also has several errors, which makes interpretations by the authors difficult to understand.

Other revisions:

85: erase s from “mitogenes”

107: missing “.”

Methods/Results: I could not quite understand the statistical techniques used for the demographic analysis. Either major details are missing, or the results (such as changes in effective population sizes) are mis-interpreted.

Paragraph starting on line 311 has multiple errors, and I am still unclear how these estimates in population size, linked to specific periods of time, can be derived from analyzing one genome. Possibly, more detail is needed about the model used - as well as caveats of limitations of such a model.

312: unclear what is meant by “the interest”

313: change to “an effective population size”

386-390: Font changes

Section missing a statement of data availability

References are formatted inconsistently and oddly, for a biological manuscript.

Throughout: the same figure legend is repeated unnecessarily in the main body of the text.

Reviewer #3: Mira-Jover et al. report a de novo reference genome assembly for the tortoise Testudo graeca, a species listed as vulnerable by the IUCN. For non-model, species of conservation concern it may be challenging to generate reference assemblies using the latest long-read sequencing technologies. I agree that even short-read based reference assemblies can be a valuable resource for a number of research questions and the authors demonstrate how using a reference guided approach can dramatically improve the contiguity of a short-read reference.

Generally, I thought the paper was well presented and the analyses sound.

My only major comment is it would be nice (perhaps in discussion) to see some discussion about the potential reference biases and other short-comings associated with using a reference guided assembly.

I cannot find the bioproject: PRJNA1086345 on NCBI. Is it embargoed?

Line 65-66: This sentence is a little confusing. Do you mean that degraded samples are not optimized for producing long contigs?

Line 82: awkward wording, please revise sentence.

Line 87: change "this source of" to These

Line 96-97: Were any of these assembled Gopherus genomes assembled using long-reads?

Table 1: Rephrase 1st sentence of caption to: Availability of chelonian reference genomes.

Table 1: Could you also included kinds of reads used for assembly and maybe a contiguity metric e.g. Contig N50?

Line 126: I don't know if I would quite call this a chromosome-level assembly given that it is based entirely on mapping to the Gopherus genome. Maybe say high quality reference genome?

Line 162: rewrite sentence: examined the expected gene completeness of our assembly with BUSCO.

Line 163: check Tetrapod spelling.

Line 183: check BRAKER spelling

Line 236: Here and in table2 Is this N50 or L50? Normally L50 is reported as an integer and N50 in base pairs.

6. PLOS authors have the option to publish the peer review history of their article (what does this mean?). If published, this will include your full peer review and any attached files.

Reviewer #1: No

Reviewer #2: No

Reviewer #3: No

---

## [Author Response · Author response to Decision Letter 0]

17 Jul 2024

Dear Dr Naser,

We wish to thank the editor and the reviewers for their constructive comments. We have revised the manuscript according to reviewers’ comments and journal requirements and have implemented the suggested changes in our revised version. We emphasize the interest of a longer, contiguous assembly through the text by better highlighting some of our results (such as BUSCO scores). We expanded a paragraph in the Discussion detailing the advantages of reference-guided assembly over using a close relative genome or a fragmented assembly. We particularly focus on aspects such as population genetics inference and genome annotation.

“Page 17, line 325: As T. graeca shows accurate differences between population and lineages, the genome herein produced is valuable for further population genomics studies. Using reference genomes from distantly related species can negatively impact SNP calling by underestimating the number of variants or biasing heterozygote calling (52). This effect is significant in turtles and tortoises (12) and, thus, obtaining a reference from the same species ensures future accurate genotyping to avoid a bias in the analyses. Highly contiguous genomes are also essential for proper gene annotation. This is clearly reflected by the drastic improvement in our BUSCO scores, with the proportion of complete single copies recovered rising from 29.4% to 95.7%. The average gene length of ca 38,000bp in Gopherus is nearly one order of magnitude higher than the scaffold N50 of 5,676bp before reference-guided assembly. At last, contiguous reference genomes are critical for accurate population genetic inference. In humans, approaches such as MSMC2 and related methods lose in accuracy for scaffold lengths under 100kb - 1Mb (53). Before reference-guided scaffolding, the longest scaffold of our assembly was 220,437 kb long, while only four scaffolds were longer than 100kb. Even assuming human-like mutation and recombination rates, which are likely higher than in tortoises, the T. graeca reference would therefore have lacked of long enough scaffolds for MSMC-like approaches, impairing demographic reconstructions. Long scaffolds are also important for genome scans of selection, which rely on the lengths of haplotypes to detect possible selective sweeps (54).”

Style requirements and Funding Statements

We have reviewed PLOS ONE style requirements (format, file naming, and references as requested in point 6 of the journal comments) and have made the necessary changes to our manuscript.

We have removed all funding-related text from the manuscript. The following Funding Statement paragraph includes all the information about financial sources:

"This work was supported by Project PID2019-105682RA-I00 and TED2021-130381B-I00, funded by the Spanish Ministry of Science and Innovation (State Research Agency) (MICIU/AEI/10.13039/501100011033), the last also with the support of the European Union 'NextGenerationEU'/PRTR. Roberto-Carlos Rodríguez Caro was supported by European Union-Next Generation EU in the Maria Zambrano Programme (ZAMBRANO 21-26)."

Regarding the Role of Funder statement, we state that the funders had no role in study design, data collection and analysis, decision to publish, or preparation of the manuscript.

We have also included the name of the ethics committee and fieldwork permission codes in the ‘Methods’ section:

“Page 8, line 128: In order to sequence the whole genome of T. graeca, we sampled a fresh road killed male tortoise from Murcia (southeastern Spain) (S1 Map of sample location). Both field sampling and the collection and treatment of biological samples were supported by the government of Murcia Region (AUF20140057) and Project Evaluation Agency of the Research Vice-rectorate of Miguel Hernandez University (Elx, Spain) (UMH.DBA.EGM.03.19).”

Data availability

During the review process, two more reference genomes were added to the NCBI database (Emydoidea blandingii, GCA_036785055.1; Graptemys geographica, GCA_037349215.1). Despite the increase in the number of chelonian reference genomes, the two sequenced species belong to freshwater turtles. Given this, we consider our work highly relevant to address the lack of genomic resources for land tortoise species. Information for the two new reference genomes is detailed in Table 1, and we have also modified the main text accordingly.

Please note that all data and the genome assembly have been submitted to NCBI under BioProject code PRJNA1086345. All scripts used for this work have also been released at https://github.com/YannBourgeois/Scripts_Genome_assembly_Tgraeca

Please find in blue all our responses to the reviewer’s comments for our manuscript. Modifications to the main text and Supplementary Information are indicated in purple below their respective comments, with page and line numbers corresponding to the new version of the manuscript.

We hope you find this new version suitable for publication in PLOS ONE.

5. Review Comments to the Author

Reviewer #1:

In this manuscript, the researchers assembled a chromosomal-level genome for an iconic land tortoise species. With only cheap short-read sequencing data, the study took advantage of the highly conserved synteny in the Chelonian group to build a reference-guided assembly.

I have one general question. The paper intends to show the advantage of using a reference genome, so I think the result should show how much improvement there is when comparing reference-guided vs. de novo assemblies. It could be the BUSCO completeness, the number of annotated genes, or the demographic reconstruction difference. Other than this, I found most of the manuscript is written clearly except in a few places that need minor editing.

We agree with the reviewer’s comments and we have added a more detailed comparison of the reference guided assembly and de novo annotation for BUSCO completeness (Table 2). We also added a paragraph explaining the advantages of this more contiguous assembly for future studies in population genomics.

Page 2 Line 51: “valuable resource to increase making direct conservation efforts” should be rephrased, maybe “valuable resource for conservation efforts.”

“Page 2, line 34: Our Testudo graeca annotated genome paves the way to delve deeper into the species’ evolutionary history and provides a valuable resource for conservation efforts on their threatened populations.”

Page 3 Line 56: to generate and sequence genomic libraries.

“Page 3, line 40: Whole genome sequencing (WGS) has become a powerful tool in evolutionary and conservation biology due to the progressive reduction of partial and economical efforts to generate and sequence genomic libraries (1).”

Page 4 Line 80: here, readers would need more background on the highly conserved synteny, like comparing Chelonians to other groups of similar age.

We took into account the reviewer suggestions and we decided to add more background of the rates of chromosomal and genome evolution in chelonians compared to some other groups.

“Page 3, line 62: The very wide diversity of ecological niches found in chelonians is not reflected in its functional diversity or genome organization, which remains highly conserved across taxa while nucleotide divergence is particularly reduced and considered a slowly-evolving group (except for mitogenomes) (11–15). High synteny is often observed in reptiles, including birds, while mammals trend to display more dynamic genomes (for a review see (16). High synteny may be facilitated by long generation times typically found in chelonians, reducing the odds of rearrangements during meiosis. The lack of recently active transposable elements (TEs) may also contribute to prevent abnormal recombination between distant copies (17).”

Page 4 Line 85: “using only mitogenomes” or “using only the mitogenomes’ tree”.

We perform this change jointly Reviewer #2 suggestion, resulting in: 

“Page 4, line 74: However, microevolutionary processes like gene flow, genomic recombination, introgression or hybridization cannot be extensively addressed using only mitogenomes trees (18).”

Page 4 Line 89: “addressing novel studies”? means “studying the origin of novel/adaptive traits”? 

We had clarified this sentence changing “novel” for “future” studies as we as we attempt to report the potential studies suggested by the authors in the cited studies.

“Page 4, line 77: These genomic resources have shed light on speciation events, ancient and recent demographic changes, and are also promising for addressing future studies, such as genomic determinants of aging, immunology, aridity tolerance or gigantism in chelonians (12–14,19–23).”

Page 6 Line 119-120: I don’t understand the sentence starting with “In general.”

We-write the sentence: 

“Page 7, line 108: Southwestern European populations have origin in North Africa (23, 25), being historically introduced on Majorca, Sardinia and the Doñana National Park (25, 26).” 

Page 6 line 124: I think there should be a sentence or two about the importance of high-quality genome for this group. 

“Page 7, line 113: Given the conservation status of the Testudo genus (along with the rest of the turtles and tortoises, with more than 50% of its species considered as Threatened) (26) and the singularity of the phylogenetic history of T. graeca throughout the Mediterranean, counting with a reference genome for this genus may contribute greatly to address new potential studies aimed to the conservation and better awareness of land tortoises.”

Page 7 Line 127: need some information on the “close relative”, like the divergence time between the two species

We thank reviewers’ suggestions and we add more information about the relationship among Gopherus and Testudo divergence by adding divergence time suggested for these two genera on Khelmaier et al., 2022 and Lourenço et al., 2013 (before cited as 47 since divergence time estimation were described on Demographic history inference, at Materials and Methods section).

Page 8, line 121: […] and we used a close relative reference available genome (G. evgoodei, approximately 50 MYA of divergence time) (18,29) to further scaffold and annotate the genome.

Page 7 Line 136: minus four degrees? Just need to confirm because usually the fridge is 4 degrees, and the freezer is -20

Here the reviewer is absolutely right. Our sample was kept frozen in our laboratory (close to -18°C). We have made the change in the manuscript. 

“Page 8, line 131: The sample was stored at -18°C.”

Page 8 Line 164: reference for the tetrapoda-0db10. Is it from the OrthoDB database?

We have checked the database source and we confirm that it is from OrthoDB database. We added a new clarification in the manuscript. 

“Page 9, line 159: We tested the continuity and the presence of 5,310 Tetrapod’s shared genes (tetrapoda_0db10, from OrthoDB database) before and after applying ntJoin and always after gap removal with GapCloser).”

Page 9 Line 177: TEs

“Page 10, line 173: We explored the age distribution of TEs by examining the divergence among the different TE families […]”

Page 13 Line 265: “Over recent (>10Mya) evolutionary times” should be “(<10Mya)”?

Page 14, line 261: […] transposable elements (TEs) over recent (<10Mya) evolutionary times. 

Page 13 Line 265: I don’t understand why the same data needs a table and a figure. Also, why does A. gigantea have nothing in the “Others”? 

We have decided to keep only a table to illustrate the distribution of repetitive elements. The figure is deleted. In addition, we have added NA to the missing data in the Table 3 (detailed in caption). This lack of information is due to the authors of the genomes described did not report the presence of all retro element families or other repetitive elements throughout the genome (others).

Table 3: Representation of the different families of repetitive elements found in the Testudo graeca assembly and other closely related available chelonians (Gopherus agassizii and Aldabrachelys gigantea). Testudo graeca and A. gigantea show similar rates of retroelements as DNA transposons over the genome, while the G. agassizii masked region results in a greater presence of retroelements, but fewer DNA transposons. We note that the G. agassizii study only ran RepeatModeler on unmasked regions, while the T. graeca and A. gigantea studies cover the whole genome. Unreported statistics are indicated as ‘NA’.

Figure 1: missing ABCD, and the legend b) should be “the number of genes annotated.”

Figure 1: Representation of the main genome assembly results.

(A) BUSCO completeness comparison with the five assembled Testudineae genomes. (B) number of genes annotated for the five Testudineae genomes and for T. graeca. (C) Circos plot of the mitogenome, with the position of the annotated genes and depth of coverage (inner circle). (D) demographic reconstruction of the large annotated chromosomes for T. graeca. 

Reviewer #2: Comments to authors:

This manuscript adds considerable information to understanding full genomes of Testudines. The only major weakness is an incomplete description of how a demographic analysis can be undertaken from a single genome. The paragraph in the Results that is related to this analysis also has several errors, which makes interpretations by the authors difficult to understand.

We now make clear in the methods section that we are realigning short reads to the reference to call variants and heterozygous sites. MSMC-like methods estimate the most recent coalescence time between haplotypes along the genome and infer a demographic history from their distribution. 

“Page 11 line 205: Historical changes in the effective population size were inferred with the MSMC2 v2.1.4 software (45). MSMC2 uses a Hidden Markov Model to estimate the most recent time since coalescence among the haplotypes under recombination. The method can be applied to a single diploid genome, but requires heterozygous sites to be identified to obtain coalescence times between the two haplotypes. To do so, we realigned the reads on the reference genome using BWA-MEM-2 (46) . We ran freebayes v1.3.2 (47) to call variants from the generated alignment file (in BAM format).”

Other revisions:

85: erase s from “mitogenes”

We tracked this change jointly Reviewer #1 suggestion, resulting in: 

“Page 4, line 74: However, microevolutionary processes like gene flow, genomic recombination, introgression or hybridization cannot be extensively addressed using only mitogenes’ trees.”

107: missing “.”

We added this “.” at the corresponding sentence. 

Methods/Results: I could not quite understand the statistical techniques used for the demographic analysis. Either major details are missing, or the results (such as changes in effective population sizes) are mis-interpreted. 

We have clarified this question in the paragraph mentioned above: 

“Page 11 line 205: Historical changes in the effective population size were inferred with the MSMC2 v2.1.4 software (45). MSMC2 uses a Hidden Markov Model to estimate the most recent time since coalescence among the haplotypes under recombination. The method can be applied to a single diploid genome, but requires heterozygous sites to be identified to obtain coalescence times between the two haplotypes. To do so, we realigned the reads on the reference genome using BWA-MEM-2 (46) . We ran freebayes v1.3.2 (47) to call variants from the generated alignment file (in BAM format).”

Paragraph starting on line 311 has multiple errors, and I am still unclear how these estimates in population size, linked to specific periods of time, can be derived from analyzing one genome. Possibly, more detail is needed about the model used - as well as caveats of limitations of such a model.

312: unclear what is meant by “the interest”

313: change to “an effective population size”

We hare rephrased these two sentences resulting in: 

“Page 16, line 299: To evaluate the applicability of T. graeca’s reference genome for demographic analyses, we reconstructed its past demographic history using MSMC2 (Fig 1d). We estimated an effective population size (Ne)”

386-390: Font changes.

We have edited the manuscript as suggested.

Section missing a statement of data availability. 

This section was submitted as a separate statement to the j

---

## [Editor Report · Decision Letter 1]

23 Jul 2024

Taking advantage of reference-guided assembly in a slowly-evolving lineage: application to Testudo graeca.

PONE-D-24-16548R1

Dear Dr. Yann Bourgeois,

We’re pleased to inform you that your manuscript has been judged scientifically suitable for publication and will be formally accepted for publication once it meets all outstanding technical requirements.

Kind regards,

Murtada D. Naser

Academic Editor

PLOS ONE
---

## [Editor Report · Acceptance letter]

1 Aug 2024

PONE-D-24-16548R1 

PLOS ONE

Dear Dr. Bourgeois, 

I'm pleased to inform you that your manuscript has been deemed suitable for publication in PLOS ONE. Congratulations! Your manuscript is now being handed over to our production team.

Kind regards, 

on behalf of

Dr. Murtada D. Naser 

Academic Editor

PLOS ONE